# Bessel Beam Dielectrics Cutting with Femtosecond Laser in GHz-Burst Mode

**DOI:** 10.3390/mi14091650

**Published:** 2023-08-22

**Authors:** Pierre Balage, Théo Guilberteau, Manon Lafargue, Guillaume Bonamis, Clemens Hönninger, John Lopez, Inka Manek-Hönninger

**Affiliations:** 1Université de Bordeaux-CNRS-CEA, CELIA UMR 5107, 33405 Talence, France; 2AMPLITUDE, Cité de la Photonique, 33600 Pessac, France

**Keywords:** Bessel beam, GHz-burst mode, laser processing, dielectrics, cutting

## Abstract

We report, for the first time to the best of our knowledge, Bessel beam dielectrics cutting with a femtosecond laser in GHz-burst mode. The non-diffractive beam shaping is based on the use of an axicon and allows for cutting glasses up to 1 mm thickness with an excellent cutting quality. Moreover, we present a comparison of the cutting results with the state-of-the-art method, consisting of short MHz-bursts of femtosecond pulses. We further illustrate the influence of the laser beam parameters such as the burst energy and the pitch between consecutive Bessel beams on the machining quality of the cutting plane and provide process windows for both regimes.

## 1. Introduction

The microcutting of transparent materials is an essential need in modern fabrication for machining of different materials with a precision on the micron scale [1]. Indeed, achieving high cutting plane finish without damaging the surface (no chipping) or subsurface while keeping high mechanical strength remains a major challenge, especially in the case of brittle materials such as glasses. Laser beam machining offers a versatile non-contact and high-precision method that potentially does not require any finishing or post-processing, thus allowing for chemical-free processes. Femtosecond laser micromachining of dielectrics represents a particularly interesting opportunity for the industry owing to the ultra-short interaction time of this technology, which enables precise and highly localized energy deposition thanks to non-linear absorption [2]. In order to meet industrial requirements in terms of productivity, innovative approaches to optimize the laser energy deposition in the material are crucial and can be improved by temporal and/or spatial beam shaping. For instance, Gaussian beam focalization suffers from a limited depth of focus in the range of few tens of microns related to the confocal parameter, restricting its use for glass cutting. However, some technical solutions using Gaussian beams remain, such as scribe and break [3]. This technique of glass cutting comprises two steps: the first step consists of generating a groove on the surface of the sample, and the cutting process is then realized by applying a mechanical stress in order to separate the two parts. This technique allows for a high processing speed but generates dust due to the ablation process and path deviation induced by important mechanical stress. Other approaches consider cutting methods such as filamentation [4], where a Gaussian beam produces an elongated energy deposition thanks to the Kerr effect within the material, multifocal cutting generated by lenses forming multiple focal points within the materials [5,6] or even the controlled fracture propagation [7,8,9,10,11,12]. The latter is slightly different and consists of creating a notch on the edge of the glass and then moving the sample in the laser beam [13]. The high temperature gradient generated by the laser induces a fracture in the sample which then follows the beam trajectory, creating a free-form defect-free cutting plane that does not require any mechanical assistance for singulation. This method, however, suffers from several drawbacks such as uncertainty in the cutting trajectory, inaccuracy due to path deviation generated by the high intensity pulses, and an overall large amount of material loss.

The use of a non-diffractive Bessel beam for microcutting offers the advantage of a dust-free and zero-kerf micromachining process. Such a Bessel beam can be produced by an axicon or a spatial light modulator and allows for an elongated and highly localized energy deposition over several millimeters thanks to its interferential character [14]. Indeed, this latter approach is already used for materials cutting [15,16,17,18], bulk modifications [19,20], nanoholes drilling, or for high-aspect ratio drilling [21,22,23,24,25]. Furthermore, this spatial beam shaping technique can be coupled with the use of MHz-bursts in order to enhance the energy deposition in a decisive way for glass cutting [15]. The principle is then to produce an intense elongated and uniform bulk modification for micro crack appearance along the glass thickness with every Bessel beam-shaped single burst. A phase mask was used to align the crack orientation along the desired cutting trajectory, which could be either straight or curved. The plane defined by subsequent micro cracks acts as pre-cutting treatment since a slight mechanical stress by hand is required to cleave and separate the pieces [20,22].

More recently, a new regime of temporal beam shaping has emerged for laser–matter interaction. The GHz-burst mode has proven its interest with Gaussian beam shaping for the ablation of glasses and sapphire [26,27,28,29,30], of copper and silicon [31,32,33], and for the drilling of crack-free high-aspect ratio holes in glasses [34]. Indeed, the pulse-to-pulse delay within the GHz-burst involves timescales in the order of magnitude of the nanosecond, which is the same as the heat relaxation time [2]. Therefore, this particular regime relies on thermal accumulation delivered by an important number of pulses within the burst at a GHz repetition rate [35]. Interestingly, the high repetition rate within the burst allows for machining materials with pulse fluences below the single pulse ablation threshold, which underlines the tremendous capacity of this regime.

In this contribution, we demonstrate, for the first time to the best of our knowledge, the GHz-burst regime coupled with Bessel beam shaping to cut several transparent dielectric materials of different thicknesses and compare it to the MHz-burst regime. In this manuscript, we considered the cutting process including both steps for practical reasons while strictly speaking we should talk about pre-cutting and cleaving. We studied different parameters such as the pitch between two consecutive Bessel beams as well as the burst energy in order to compare the results obtained in both MHz- and GHz-burst regimes. The overall cutting plane finish quality was studied in terms of surface roughness as well as the trajectory tracking with comparable parameters. Furthermore, we investigated a larger set of parameters in order to determine the processing windows for both regimes. Indeed, these two regimes reveal working with rather different parameters.

## 2. Experimental Setup

### 2.1. Laser System

The used laser system for this experiment is a Tangor 100 from Amplitude emitting pulses at 1030 nm with a pulse duration around 500 fs. The output beam is linearly polarized. This versatile laser system allows us to work both in MHz-burst regime with 4 pulses per burst at 40 MHz and in GHz-burst regime with 50 pulses per burst at a repetition rate of 1.28 GHz. The burst repetition rate is variable up to 200 kHz. We chose to work with bursts of constant intensity, meaning that the energy is evenly distributed among the pulses, as described in [36], as a flat burst.

### 2.2. Micromachining Workstation

The Bessel beam shaping module was mounted on a *Z*-axis motorized stage (VP25X, MKS Instruments) which allowed for a precise positioning of the Bessel beam in the sample. The latter was fixed on XY-monolithic motorized stages (One-XY60, MKS Instruments) for positioning the sample under the laser beam and for setting the translation velocity during the cutting process. The ratio between velocity and inter burst repetition rate defines the pitch between two subsequent bursts into the sample. In order to optimize this process, our station was equipped with a side-view imaging system composed of a green diode emitting at 523 nm and a Basler CMOS camera (Basler acA1920-25mu, 1/3.7″ sensor, resolution of 1920 × 1080 p, pixel size 2.2 µm × 2.2 µm, 25 i/s, rolling shutter) coupled with a long-distance microscope (InfiniMax KX with MX5 objective) with a 520 nm bandpass filter in order to visualize directly horizontally through the samples and not being blinded by the processing laser wavelength. The whole machining station was mounted on a granite base and gantry in order to ensure the stability and the repeatability of the experiments. The translation stages, as well as the laser gate, were controlled by the DMCpro software (Direct Machining Control, Vilnius, Lithuania) which allows us to have a perfect mastery of the machining station and the cutting parameters.

### 2.3. Generation and Characterization of the Bessel Beam

The spatial beam shaping setup to generate the Bessel beam, inspired from [15], is represented in Figure 1 and uses an axicon with an apex of 170°. The axicon creates a long and wide primary Bessel beam which is then imaged into a secondary Bessel beam using a set of lenses in order to reduce the dimensions and maximize the deposited energy in a smaller volume.

From a theoretical point of view, the parameters of the primary Bessel beam can be calculated in air with the following formulas [15,37,38]:(1)$$\theta_{1}=(n_{a}-1)\times \alpha$$
(2)$$\alpha =90-\frac{apex}{2}$$
with *n_a_* the optical index of the axicon, *α* the base angle of the axicon, and *θ*_1_ the half-angle of the primary Bessel beam, as indicated in Figure 1. For small *θ*_1_, the length *l* and the central core radius *r*_1_ of the primary Bessel beam at the zero-intensity point are given by [15,37,38,39]:(3)$$l=\frac{d}{{2\theta }_{1}}$$
(4)$$r_{1}=\frac{2.4\times \lambda }{2\pi \sin\left(\theta_{1}\right)}$$
where *d* is the beam diameter when entering the axicon and *λ* is the laser wavelength. The secondary Bessel beam parameters can be determined using the primary Bessel beam parameters and the magnification coefficient given by *M* = *f*_2_/*f*_1_ with *f*_1_ and *f*_2_ the focal lengths of the two lenses, respectively, in this case *f*_1_ = 125 mm and *f*_2_ = 10 mm. The secondary Bessel beam half-angle *θ*_2_, length *L*, and radius *r*_2_ must be calculated in the glass sample and are given by the following formulas [15,37,38]:(5)$$\theta_{2}=\frac{\theta_{1}}{M\times n_{g}}$$
(6)$$L=\frac{Md}{{2\theta }_{2}}$$
(7)$$r_{2}=M\times r_{1}$$
with *n_g_* the refractive index of the glass sample. Note that the glass refractive index theoretically has no impact on the diameter of the secondary Bessel beam. From a practical point of view, the length of the Bessel beam can be adjusted by changing the beam size of the beam upstream the axicon and the radius by modifying the magnification parameter *M*. These formulas give the following results: the calculated primary Bessel beam half-angle is *θ*_1,th_ = 2.3°, the radius of the primary Bessel beam is *r*_1,th_ = 9.8 µm, the half-angle of the secondary Bessel beam is *θ*_2,th_ = 28.6°, and the theoretical radius of the secondary Bessel is calculated as *r*_2,th_ = 0.8 µm. However, the simple approach of geometrical optics is not suited for calculations on the order of magnitude of the wavelength, and thus, this theoretical value indicates only a lower limit [14]. We therefore performed careful measurements of the primary and secondary Bessel beams, respectively, and a schematic drawing of the primary Bessel beam measurement is depicted in Figure 2.

In this figure, P1, P2, and P3 are acquisition planes for the measurement of the primary Bessel beam radius *r*_1,exp_ and the primary Bessel beam half-angle *θ*_1,exp_, respectively. Figure 3 displays the corresponding beam analyzer (WinCamD-XHR, pixel size 3.2 µm) acquisitions. Figure 3a displays the intensity profile corresponding to P1 that we used to measure the diameter of the primary Bessel beam. We adapted the beam size to the beam analyzer resolution to ensure a reliable measurement using a homemade 10 X-magnification system that was carefully calibrated with an accuracy of the magnification factor of about 2%. Figure 3b,c correspond to the intensity profiles acquired in planes P2 and P3 for the measurement of the angle, where the distance H between P2 and P3 was set to 1 mm. From Figure 3b,c we can determine the half-angle of the Bessel beam with the following formula by extracting the diameters of the circles:(8)$$\theta_{1,exp}=arctan\left(\frac{D-d}{2H}\right)$$
with *D* and *d* being the diameters of the circles, respectively, and *H* being the distance between the two acquired planes. In our case, we obtained *θ*_1,exp_ = 2.34° with an uncertainty of ±0.01° and a radius *r*_1,exp_ = 9.00 µm with an uncertainty of ±0.66 µm. In addition, we measured the secondary Bessel beam half-angle by exactly the same method and we obtained *θ*_2,exp_ = 27.26° with an uncertainty of ±0.01° and a secondary Bessel beam radius *r*_2,exp_ = 1.9 µm with an uncertainty of ±0.66 µm. The experimental values for the primary Bessel beam half-angle and radius, as well as the secondary Bessel beam half-angle, are very close to the theoretically calculated ones. Only the secondary Bessel beam radius appears to differ, but that could be explained by the fact that the dimensions are approaching the diffraction limit.

Additionally, we estimated the length of the Bessel beam by visualizing the modification in a glass sample, as can be seen in Figure 4a, with the luminescence during the interaction (top) and the resulting bulk modification (bottom). The resulting length was estimated to be 1.2 mm with an uncertainty of ±0.2 mm compared to a calculated one of 1.8 mm. The difference here arises from the fact that the real length of the Bessel beam is probably higher than the resulting measured bulk modification.

The use of an axicon in our setup imposes a preliminary step before cutting. Indeed, due to imperfections in the tip of the axicon, a crack with a random orientation appears following the energy deposition. The orientation of the crack can be controlled either by tilting the axicon itself [19] or by rotating a phase mask which is placed upstream the axicon [40]. In our case, we chose to implement the second configuration and to orientate the crack with a phase mask as illustrated in Figure 4b in a sodalime sample. On Figure 4b, each dot results from a single burst, and the distance between two dots defines the pitch. The goal here is to align the crack with the trajectory of the cut. In the case represented here, the right orientation is shown in green with vertical cracks. In red we can see the most detrimental orientation of the crack, which could lead to a lowered surface quality during the cut.

### 2.4. Roughness and Topography Measurements

Both topography and surface roughness (Sa) measurements of the sidewalls (i.e., the cutting planes) were performed using a confocal profilometer (Smart Proof 5, ZEISS) equipped with a 20X objective with a numerical aperture of 0.7. For samples exceeding a thickness of 500 µm, the sidewalls are wider than the measurement field of the confocal microscope, so the surface roughness Sa was estimated as an average on several measurements covering the full sidewall.

### 2.5. Samples

The cutting experiments were carried out on samples of 1 mm-thick sodalime, on 200 µm-thick fused silica, 300 µm-thick alkali free borosilicate (AF32, SCHOTT Glass, Perai, Malaysia) glass, and 430 µm-thick sapphire. Fused silica and sapphire have a similar bandgap (about 9 eV [41]), which is higher than the one of sodalime (3–4 eV [42]). Furthermore, sapphire has a much higher thermal conductivity (about 40 W/m/K [43]) compared to fused silica (1.81 W/m/K [44]), AF32 glass (1.5 W/m/K determined by the flying spot method [45]), and sodalime glass (1.42 W/m/K [44]).

## 3. Cutting Results and Discussion

In this section, we present the experimental results as well as the corresponding discussions. We divided this study into four subsections. The first one is exclusively dedicated to the GHz-burst Bessel beam cutting with a study of the surface roughness as a function of the pitch for several burst energies. The second section is a comparative study between Bessel beam cutting in the MHz-burst and the GHz-burst regimes for comparable burst energies and pitches. These first results reveal two main points. Firstly, we noticed that the two regimes operate with very different parameters which lead us to investigate a wider range of parameters in order to extract the processing windows of both regimes, which is the object of the third subsection. Secondly, we observed an over-exposure regime appearing for low pitches and/or high burst energies. The fourth subsection is dedicated to this regime.

### 3.1. GHz Burst Bessel Beam Dielectrics Cutting

This first part is dedicated to GHz-burst Bessel beam cutting of sodalime glass, fused silica, and sapphire for different burst energy and pitch values. The cutting quality was evaluated regarding the surface roughness (Sa) of the sidewalls (cutting planes) after singulation. We first chose to investigate the behavior of the GHz-burst Bessel beam cutting regime itself, as it has been completely unknown up until now. To do so, we studied the evolution of the surface roughness (Sa) of the cutting plane as a function of the pitch between two consecutive Bessel beams within the materials for different burst energies with bursts of 50 pulses at a repetition rate of 1.28 GHz. Figure 5b,d,f depict the results for the three materials applying four different burst energy values of 253, 294, 337, and 383 µJ, respectively. The error bars on these graphs correspond to the uncertainty on the measurements which was estimated to be ±10% of the surface roughness value. Indeed, for each acquisition the resolution of the profilometer depends on the scanned range, as well as on the number of acquired planes.

The results are similar for sodalime (b) and fused silica (d). For high pitch values, the surface roughness is rather high, as the Bessel beams were too far apart from each other and the mechanical stress that had to be applied was important, leading to a lowered surface quality. By diminishing the pitch, the surface quality increases (i.e., decreasing Sa values), reaching a minimum value which depends on the materials itself as well as on the burst energy. Following that, for pitches that were too low, the surface roughness increases again, evidence of a too-high overlap and a thermal load that is too important for the material to support. The points corresponding to this over-exposure do not appear on this graph as the cutting plane is not uniform at all, either concave or convex, and the corresponding roughness measurements are neither reliable nor relevant. This point will be further investigated in Section 3.4. In contrary to the glasses, there is no obvious trend that can be deduced from the results depicted in the graph concerning sapphire (Figure 5f).

The surface morphologies corresponding to the lowest surface roughness of the cutting planes after singulation are shown in Figure 5a,c,e. These results were obtained for a burst energy of 253 µJ and a pitch of 0.04 µm in sodalime, a burst energy of 294 µJ and a pitch of 0.1 µm in fused silica, and a burst energy of 337 µJ and a pitch of 0.04 µm in sapphire, respectively. The cutting quality is very good, since the cutting planes are regular and smooth in both sodalime and in fused silica where the surface roughness Sa is 0.47 µm and 0.75 µm, respectively, and there is no path deviation. The cutting plane in sapphire is less uniform and exhibits a higher surface roughness (1.17 µm). Some hilly, uncracked, and glossy areas reveal that the pre-cutting has been incomplete, and thus the mechanical stress required for singulation increased for sapphire. This observation can be explained by the high bandgap value, and especially the high thermal conductivity of sapphire (about 40 times higher compared to the investigated glasses). Indeed, the cutting process in GHz-burst mode is more sensitive to the bandgap value since the low pulse energy of the individual pulses within a burst produces less non-linear absorption. This behavior remains, even for relatively high burst energies. Nevertheless, we observed no path deviation.

Furthermore, we noticed that there is no visible effect and no pre-cutting effect in fused silica or in sapphire for the lowest burst energy value (253 µJ), whereas it is possible to pre-cut sodalime glass in these conditions. We assumed that this is due to the higher bandgap value for fused silica and sapphire compared to sodalime. Moreover, in sodalime and fused silica, pre-cutting can be achieved with pitches as high as 2 µm. Meanwhile, for sapphire, it is not the case for pitches exceeding 0.1 µm. We assumed that this can be explained by the higher thermal conductivity of sapphire compared to sodalime and fused silica, where the higher thermal conductivity prevents from pre-cutting with higher pitches. On the basis of this observation, we deduced that there is a slight heat accumulation between bursts. In the case of glasses, the material does not fully relax between two subsequent bursts. Thus, the pre-cutting process takes advantage of a low thermal conductivity value thanks to a beneficial spatial cooperative effect between subsequent bursts.

### 3.2. Comparison between MHz-Burst and GHz-Burst Cutting

We performed a first comparative study between MHz-burst and GHz-burst cutting on sodalime and on AF32 glass (alkali-free alumino-borosilicate glass from Schott). We kept the same experimental conditions for both regimes and investigated two burst energies, 200 µJ and 215 µJ, in a pitch range from 0.005 µm to 2 µm with bursts of four pulses at 40 MHz repetition rate and 50 pulses at 1.28 GHz repetition rate. The graphic representation of the surface roughness as a function of the pitch and the profilometer images of the best results for the two materials, 1 mm-thick sodalime and 300 µm-thick AF32, are displayed in Figure 6. For the reader’s convenience, the data are summarized in Table 1.

The images corresponding the GHz-burst regime (on the left) in sodalime show partly a very regular and homogenous surface that has been cut by the Bessel beam, and we partly observe a cleaved plane. This observation can be explained by the fact that the pulses within the burst in this regime carry a low energy which highly reduces the effective length of the Bessel beam. Obviously, the chosen burst energies are too low to generate a long effective Bessel beam. Contrarily, for AF32, we observe regular cutting planes with low surface roughness values in the GHz-burst regime. However, on the pictures on the right corresponding to MHz-burst cutting, one may notice irregularities and even bubble-like forms on AF32. These particular structures may attest to a too-high burst of energy for the MHz-burst regime for which each pulse within the burst carries almost 13 times more energy than the pulses within a GHz-burst. For high pitch values, the surface roughness is high, especially in the GHz-burst mode. Indeed, in this particular regime, the energy deposition is much more localized in the material, and therefore, a pitch that is too high (i.e., an overlap that is too low) does not allow for generating a continuous cutting plane but rather represents a succession of Bessel beam-induced cracks (Figure 4b), resulting in a high surface roughness. When diminishing the pitch, there seems to be an optimum value for which the surface roughness reaches a minimum. Following that, for too low pitch values, the surface roughness increases again, especially in the MHz-burst mode. Indeed, as the pulses carry much more energy, an overlap that is too high can lead even sooner to an over-exposure which is detrimental for the surface quality.

### 3.3. Optimization of the Parameters for Both Regimes

We extended the range of processing parameters concerning burst energies and pitches to independently identify the best sets of parameters for both MHz- and GHz-burst regimes in terms of cutting quality. We varied the burst energy from 147 to 383 µJ, and the pitch from 0.005 to 10 µm. The bests results obtained in both materials for both regimes are displayed in Figure 7. Note that both regimes produced an excellent cutting quality with very regular cutting planes, especially in AF32. In GHz-burst mode, the resulting Sa of the cutting plane is as small as 0.27 µm (Figure 7d). Moreover, with the adapted burst energy, it was then possible to obtain a full cutting plane over the whole sample area on 1 mm-thick sodalime, even in GHz-burst mode.

This figure demonstrates that the parameters have to be chosen differently in MHz-burst and GHz-burst mode in order to obtain the best cutting plane qualities. Given that the energy is spread within 50 pulses in the GHz-burst mode, it is necessary to have a burst energy that is high enough to reach non-linear absorption along the whole glass thickness. Moreover, an important overlap is also needed to take advantage of the heat accumulation that drives the GHz-burst cutting process. Regarding the MHz-burst mode, the pulses are much more energetic, therefore a lower burst energy and less overlap can be used for cutting. However, we can see that even for the appropriate set of parameters, the best surface quality (i.e., lower surface roughness) is still obtained in GHz-burst mode.

We investigated the processing windows for GHz-burst and MHz-burst cutting for sodalime, and the results are represented in Figure 8. By heuristic considerations we identified four zones depending on burst energy and pitch value. Note that the proportions and the shapes of the windows represented here are indicative and might, in detail, depend on other experimental conditions. The black crosses in these graphs were extracted from the experimental data acquired on sodalime. For the sake of clarity, we have not represented all the experimental points, as they are sometimes very close to each other. We observed a lower limit for the burst energy in both regimes—around 180 µJ in GHz-burst mode and 100 µJ in MHz-burst mode.

The green zone (1) corresponds to the optimum process window producing the best cutting results in terms of cutting feasibility, uniformity, and surface roughness. Its para-meters are very different in the GHz-burst regime with respect to the MHz-burst regime. The former requires relatively high burst energy and a low pitch between consecutive Bessel beams, while the latter requires a low burst energy but a higher pitch. The orange zone (2) represents a processing window for which cutting is possible but with a lowered surface quality of the cutting plane due to a too-high overlap (low pitch). The red zone (3) represents the parameters for which cutting is not feasible due to insufficient bulk modification. The mechanical stress required for singulation is too high, resulting in an uncontrolled breaking of the sample. In this range of pitches in MHz-burst mode, the third window is only visible in the lower part of the graph, corresponding to very low burst energy as singulation was possible as soon as there was a visible plane. The grey zone (4) corresponds to very low pitches and/or too high burst energy, leading to the thermal cutting regime which will be further discussed in the next section. Despite the fact that no mechanical stress is required in this operating window, the cutting plane no longer follows the Bessel beam trajectory, which can be critical in industrial applications.

### 3.4. Over-Exposure

As indicated in the former sections, for an overlap that is overly important in both MHz- and GHz-burst configurations, depending on the burst energy, we observed that a thermal cutting regime can be reached. In this regime, applying stress to cleave the sample was not necessary and the cut part fell off itself. However, we observed that the cutting plane did not follow the trajectory defined by the Bessel beam. Indeed, for these configurations, the thermal load is too high with respect to the heat dissipation, leading the material to show a large heat-affected zone (HAZ), eventually reaching the softening temperature of the material [46]. The cutting plane will then follow the border of the heat-affected zone, resulting in a path deviation both from the trajectory and from the straightness of the Bessel beam along the sample thickness, as illustrated in Figure 9. The resulting cutting plane is curvilinear, either concave or convex, as can be seen in the 3D representation of the surface taken with the profilometer (Figure 9, bottom).

On the top of this figure, a schematic drawing of the HAZ is depicted. On the microscope image (middle), we observe that there are points of energy concentration such as those in Figure 9 of Ref. [38]. Furthermore, we noticed that detrimental extended cracks appear along the Bessel beam due to heat accumulation when the burst repetition rate is too high. A similar phenomenon has been observed in drilling experiments in the GHz-burst regime [47]. Indeed, in sodalime, for example, the burst repetition rate in the drilling experiments was limited to 10 kHz, while cutting with the Bessel beam is possible with repetition rates up to 20 kHz without HAZ appearing. This can be explained by the fact that the volume in which the energy is deposited is much larger for a Bessel beam than in percussion drilling using a Gaussian beam. In the case represented here, a large HAZ is displayed.

## 4. Conclusions

In this study, femtosecond laser GHz-burst dielectrics cutting with a Bessel beam shaping was demonstrated for the first time to the best of our knowledge, and the results were compared to the MHz-burst regime. We revealed that the operating windows of each regime are very different in terms of burst energies and pitches to achieve a clear cut. However, we observed a thermal cutting regime that can prove detrimental in terms of path deviation in both axes, and we indicate how to avoid this regime. Furthermore, we achieved excellent cutting plane qualities in both burst-mode regimes. However, for optimized parameters, the best quality with the lowest surface roughness was systematically obtained in the GHz-burst regime, attesting that this combination of spatial and temporal beam shaping can lead to a better-controlled cutting process. The comparison of the cutting results obtained with both GHz- and MHz-bursts of femtosecond pulses proved that the GHz-burst regime is perfectly suited for high quality micromachining of dielectrics with minimum surface roughness Sa of the cutting planes.

## Figures and Tables

**Figure 1 micromachines-14-01650-f001:**
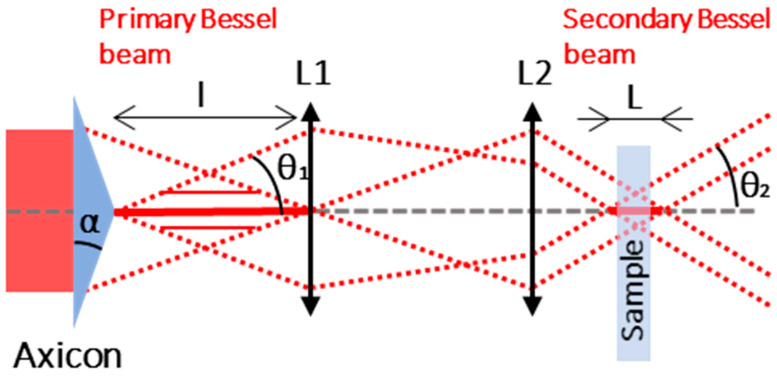
Schematic of the Bessel beam generation using an axicon and a set of lenses.

**Figure 2 micromachines-14-01650-f002:**
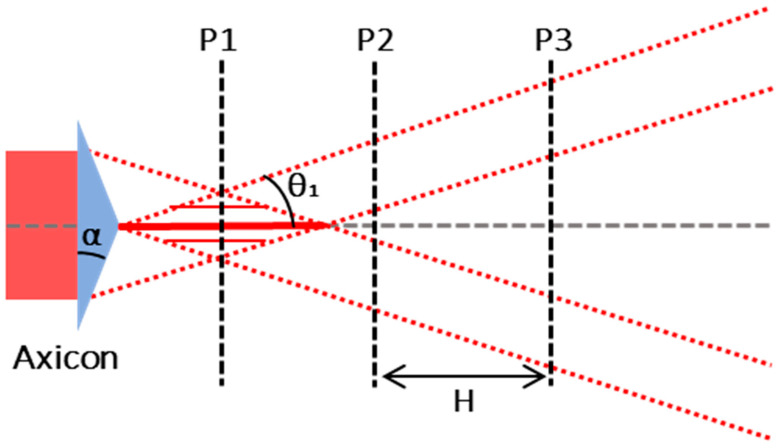
Schematic of the measurement method for the primary Bessel beam.

**Figure 3 micromachines-14-01650-f003:**
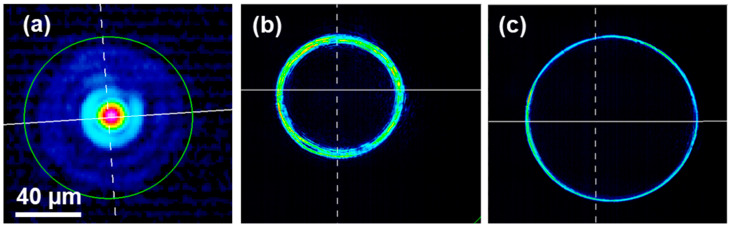
Measurement of the primary Bessel beam diameter (**a**). Intensity profile measured at two different planes (**b**,**c**). All measurements are made using a WinCamD beam analyzer.

**Figure 4 micromachines-14-01650-f004:**
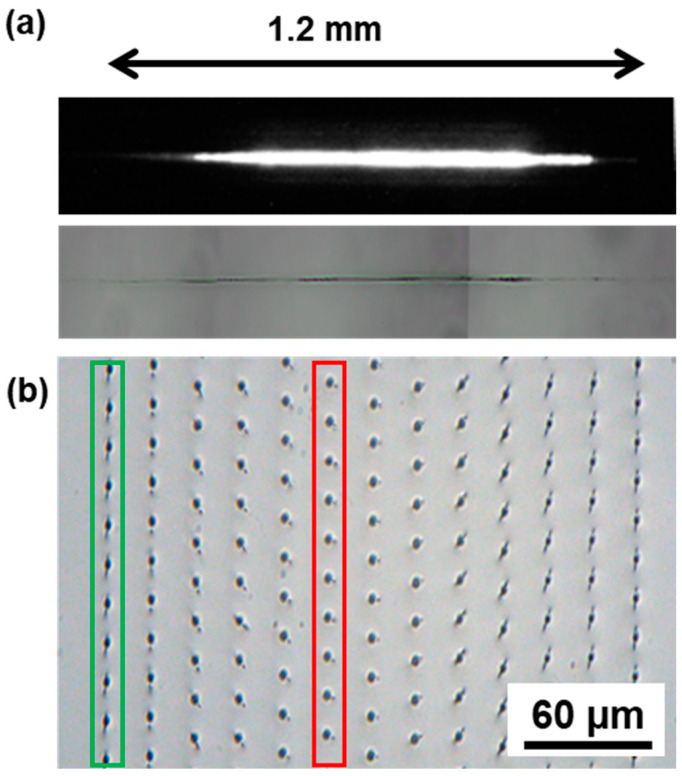
Characterization of the Bessel beam by sideview imaging of the luminescence of the Bessel beam and of a laser induced modification in a glass sample. (**a**) Top view image of the crack orientation using a phase mask upstream the axicon with a pitch of 20 µm (**b**).

**Figure 5 micromachines-14-01650-f005:**
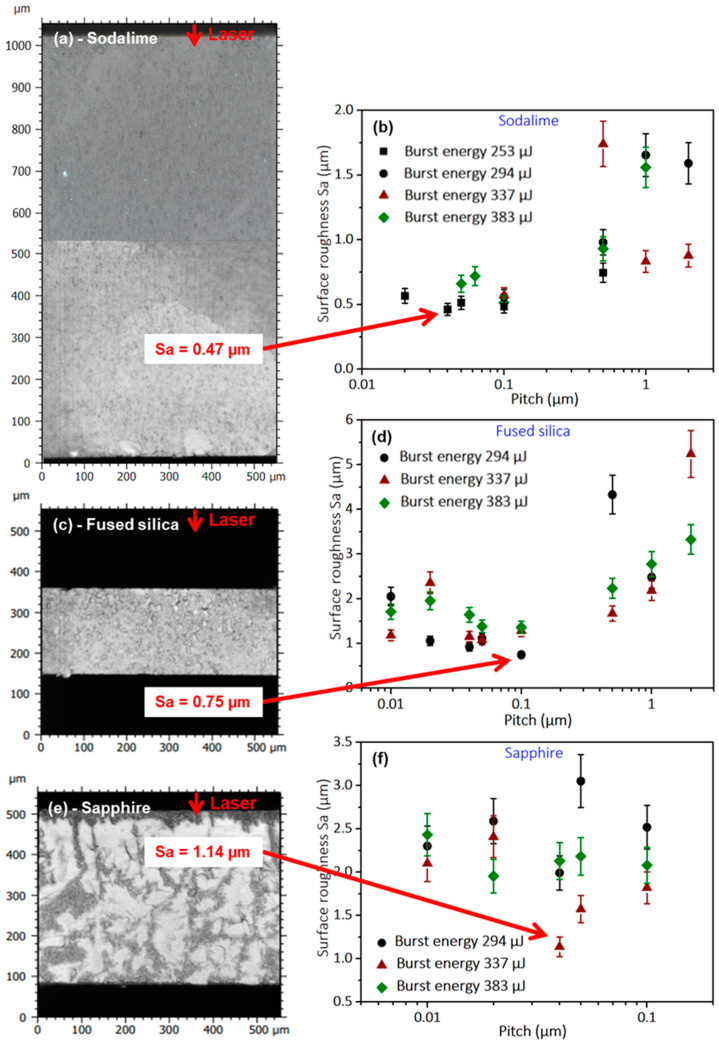
Topography measurements on sidewalls after singulation corresponding to the lowest surface roughness obtained in our study in 1 mm-thick sodalime for a burst energy of 253 µJ and a pitch of 0.04 µm for a resulting Sa of 0.47 µm (**a**), in 200 µm-thick fused silica for a burst energy of 294 µJ and a pitch of 0.1 µm for a resulting Sa of 0.75 µm (**c**), and in 430 µm-thick sapphire for a burst energy of 337 µJ and a pitch of 0.04 µm for a resulting Sa of 1.14 µm (**e**). Surface roughness as a function of the pitch in sodalime for burst energies in a range from 253 µJ to 383 µJ (**b**), in fused silica for burst energies in a range from 294 µJ to 383 µJ (**d**), and in sapphire for burst energies in a range from 294 µJ to 383 µJ (**f**). Note the pitch scale differences, especially in sapphire. Laser comes from the top.

**Figure 6 micromachines-14-01650-f006:**
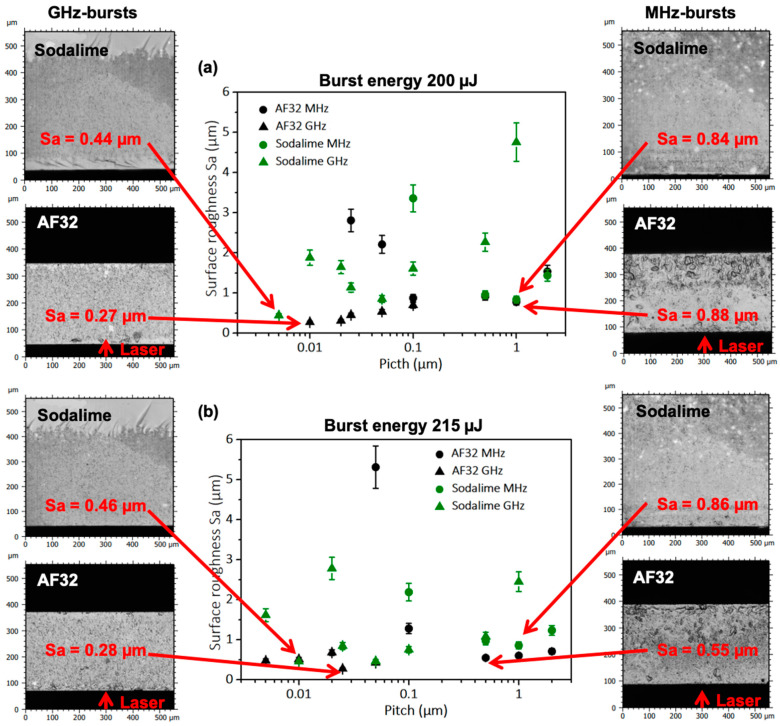
Graphic representation of the surface roughness as a function of the pitch between two consecutive Bessel beams in sodalime and AF32 for burst energies of 200 µJ (**a**) and 215 µJ (**b**). The images corresponding to the bests results obtained are displayed on the left for GHz-bursts and on the right for MHz-bursts. Laser comes from the bottom.

**Figure 7 micromachines-14-01650-f007:**
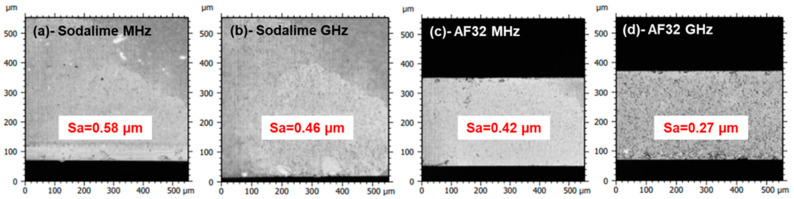
Profilometer images of the results obtained in sodalime for a pitch of 1 µm in MHz-burst mode for a burst energy of 147 µJ with a resulting Sa of 0.58 µm (**a**), in sodalime for a pitch of 0.04 µm in GHz-burst mode for a burst energy of 194 µJ with a resulting Sa of 0.46 µm (**b**), in AF32 for a pitch of 1 µm in MHz-burst mode for a burst energy of 127 µJ with a resulting Sa of 0.42 µm (**c**), obtained in AF32 for a pitch of 0.025 µm in GHz-burst mode for a burst energy of 194 µJ with a resulting Sa of 0.27 µm (**d**).

**Figure 8 micromachines-14-01650-f008:**
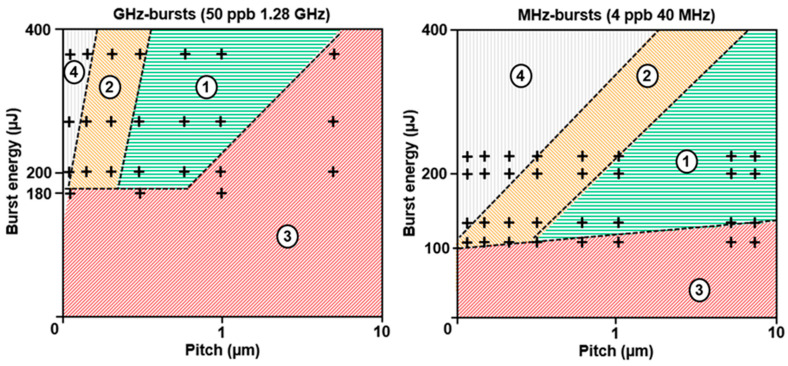
Schematic representation of the different operating windows that appeared during the cutting study. The values of burst energies, pitch, and crosses correspond to experimental data in sodalime. The green zone (1) corresponds to the optimum process window, the orange zone (2) represents a process window for which cutting is possible but with a lowered surface quality of the cutting plane, the red zone (3) represents the parameters for which cutting is not feasible, and the grey zone (4) corresponds to very low pitches and/or too high burst energy leading to the thermal cutting regime.

**Figure 9 micromachines-14-01650-f009:**
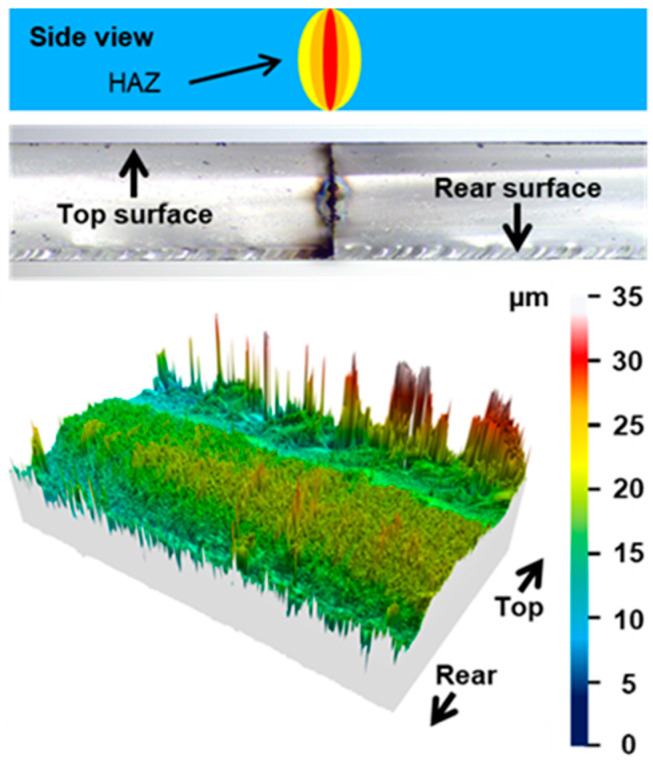
Schematic view of the HAZ appearing with high overlapping (**top**) with the corresponding microscope sideview image of the cutting plane (**middle**), and 3D representation of the surface measured with the profilometer (**bottom**).

**Table 1 micromachines-14-01650-t001:** Summary of the best results obtained during this comparative study for burst energies of 200 µJ and 215 µJ for sodalime and AF32, respectively, with the corresponding pitches.

	Sodalime				AF32			
	MHz Bursts		GHz Bursts		MHz Bursts		GHz Bursts	
Burst Energy (µJ)	Pitch (µm)	Sa (µm)	Pitch (µm)	Sa (µm)	Pitch (µm)	Sa (µm)	Pitch (µm)	Sa (µm)
200	1	0.84	0.005	0.44	1	0.88	0.01	0.27
215	1	0.86	0.01	0.46	0.5	0.55	0.025	0.28

## Data Availability

Data underlying the results presented in this paper are not publicly available at this time, but may be obtained from the authors upon reasonable request.

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
