# Peer review of "Bessel Beam Dielectrics Cutting with Femtosecond Laser in GHz-Burst Mode"

_micromachines, 2023, doi:10.3390/mi14091650_

Round 1

Reviewer 1 Report

1.       Eq.(3): as I can see from the geometry of Fig.1, l=d/(2 sin θ1). It means that expression (3) is valid for small θ1 only. At the same time, the time, the Eq.(4) is written without assuming that is written without assuming that θ1 is small.

2.       Linе 166: 10X-magnification system that was carefully calibrated at 2%. Please clarify what precisely measures 2% and of which quantity?

3.       Line 445-446. Which criteria were chosen to identify four burst/pitch zones? Was it some clusterization algorithm or some euristic considerations?

4.       Please check the author(s) spelling in Ref.5

Author Response

Dear reviewer,

We would like to thank you very much for your careful reading of our manuscript, your very positive feedback and your valuable questions and suggestions. Please, see our answers in the following:

1.       Eq.(3): as I can see from the geometry of Fig.1, l=d/(2 sin θ1). It means that expression (3) is valid for small θ1 only. At the same time, the time, the Eq.(4) is written without assuming that is written without assuming that θ1 is small.

Our response: We fully agree and thank you very much for the remark. We added “for θ1 small” in line 125 just before Eq. (3).

2.       Linе 166: 10X-magnification system that was carefully calibrated at 2%. Please clarify what precisely measures 2% and of which quantity?

Our response: Thank you very much for your careful reading. Indeed, the sentence misses some words to clarify. We change into:

“…10X-magnification system that was carefully calibrated with an accuracy of the magnification factor of about 2%.”

3.       Line 445-446. Which criteria were chosen to identify four burst/pitch zones? Was it some clusterization algorithm or some euristic considerations?

Our response: Thank you very much for the question. Indeed, no algorithm was applied, the result is based on heuristic considerations. We added this and change into:

“By heuristic considerations we identified four zones depending on burst energy and pitch value.”

4.       Please check the author(s) spelling in Ref.5

Our response: Thank you very much for your particular attention. We corrected the names and authors spelling in Ref. 5.

With best regards and on behalf of all the authors,

Inka Manek-Hönninger

Reviewer 2 Report

The authors reported a Bessel beam dielectrics cutting with a femtosecond laser in GHz-burst mode and illustrated the influence of the laser beam parameters such as the burst energy and the pitch between consecutive Bessel beams on the machining quality of the cutting plane and provide process windows for both regimes. However, the authors haven’t show sufficient supporting for clarifying the topic of the manuscript. In order to support its publication in Micromachines, the authors should also address the following points, as listed.

1. In figure 4b, we can clearly find a variation of the laser modified dots, is that from the polarization dependence within the setup?

2. The GHz burst is known to be better than the MHz burst for laser machining, also as shown by the authors. To further support the conclusions in the manuscript, the authors should provide a thermodynamic model for comparing the nanosecond and the microsecond pulses.

3. In table1, a 5nm pitch is even used, the authors should clarify the meaning of using the pitch size of 5nm while the surface roughness may reach 0.46um.

4. Surface modification with fast pulsed lasers is booming in the recent years. Related references could be cited in the reference list.

a. Resonant Laser Printing of Optical Metasurfaces. March 2022 Nano Letters 22(7)

DOI: 10.1021/acs.nanolett.1c04874

b. High efficiency femtosecond laser ablation with gigahertz level bursts. Journal of Laser Applications, 2019. 31(2): p. 022205.

 Minor editing of English language required

Author Response

Dear reviewer,

We would like to thank you very much for your careful reading of our manuscript and your valuable suggestions for its improvement. 

1. In figure 4b, we can clearly find a variation of the laser modified dots, is that from the polarization dependence within the setup?

Our response: Thank you very much for the question. The crack orientation is modified by turning the phase mask (and not by turning the polarization). We might not have been clear in this point. We change into:

“The orientation of the crack can be controlled either by tilting the axicon itself [19] or by rotating a phase mask which is placed upstream the axicon [40].”

2. The GHz burst is known to be better than the MHz burst for laser machining, also as shown by the authors. To further support the conclusions in the manuscript, the authors should provide a thermodynamic model for comparing the nanosecond and the microsecond pulses.

Our response: Thank you for the suggestion which would be perfectly adapted for a review paper comparing also cutting with nanosecond and microsecond pulses. However, our paper is solely dealing with femtosecond pulses. Therefore, we consider that this would be out of the scope of this work.

3. In table1, a 5nm pitch is even used, the authors should clarify the meaning of using the pitch size of 5nm while the surface roughness may reach 0.46um.

Our response: Thank you for your remark. Maybe, we should have clarified that the pitch, which is the distance of consecutive Bessel beams, is given by the repetition rate and the speed of sample translation. The reviewer might be more familiar with the expression of “overlap” as often used for surface modifications. For cutting, one prefers using the term “pitch” as there is not necessarily an overlap. We added the information in line 219:

“On Fig. 4 (b), each dot results from a single burst, and the distance between two dots defines the pitch.”

4. Surface modification with fast pulsed lasers is booming in the recent years. Related references could be cited in the reference list.

a. Resonant Laser Printing of Optical Metasurfaces. March 2022 Nano Letters 22(7)

DOI: 10.1021/acs.nanolett.1c04874

b. High efficiency femtosecond laser ablation with gigahertz level bursts. Journal of Laser Applications, 2019. 31(2): p. 022205.

Our response: Thank you very much for the suggestions of adding further references. However, surface modifications in the sense of your first suggestion are not subject of this work, and the second suggestion is one of our own papers. So, we prefer not to add these two references to avoid dispersion and to limit self-citations.

We checked the English editing, and some minor typos were corrected. 

Best regards on behalf of all the authors,

Inka Manek-Hönninger